# Access or Accessibility? A Critique of the Urban Transport SDG Indicator

**Mark Brussel [1],\*** , **Mark Zuidgeest [2]**, **Karin Pfeffer [1]** and **Martin van Maarseveen [1]**

[1]  Faculty of Geo-information Science and Earth Observation, University of Twente, PO box 217,
7500 AE Enschede, The Netherlands; k.pfeffer@utwente.nl (K.P.);
m.f.a.m.vanmaarseveen@utwente.nl (M.v.M.)

[2]  Faculty of Engineering & the Built Environment, University of Cape Town, Private Bag X3, Rondebosch,
7701 Cape Town, South Africa; mark.zuidgeest@uct.ac.za

\*  Correspondence: m.j.g.brussel@utwente.nl; Tel.: +31-53-487-4497

**Abstract:** Progress towards the UN Sustainable Development Goals (SDGs) is being evaluated through the use of indicators. Despite the importance of these indicators, the academic community has done little in terms of a critical reflection on their choice, relevance, framing and operationalization. This holds for many SDG domains, also for the urban sector domain of target 11. To partially address this void, we aim to critically review the UN methodology for the urban access indicator, SDG indicator 11.2. In discussing its conceptual framing against the background of paradigm shifts in transportation planning, we argue that this indicator has a number of shortcomings. The most important one is that it is supply oriented and measures access to transportation infrastructure, rather than accessibility to activity locations. As an alternative, we develop two accessibility indicators that show substantial variation in accessibility across geographical areas. We implement all indicators for the city of Bogotá in Colombia, using a geo-information based approach. Our results show that SDG indicator 11.2 fails to represent the transport reality well. Its supply oriented focus neglects transport demand, oversimplifies the transport system and hides existing inequalities. Moreover, it does not provide useful evidence for targeting new interventions. The proposed accessibility indicators provide a more diverse, complete and realistic picture of the performance of the transport system. These indicators also capture the large spatial and socio-economic inequalities and can help to target improvements in urban transportation.

**Keywords:** SDG indicator 11.2; urban transport; accessibility; public transport; Bogotá

---

## 1. Introduction

### 1.1. The UN Sustainable Development Goals (SDGs) and Transportation

The United Nations (UN) launched the Sustainable Development Goals (SDGs)—a global initiative to end poverty, fight inequality and tackle climate change, in September 2015. The SDGs consist of 17 goals, ranging from general goals on poverty and hunger to more specific goals on strong institutions and partnerships [1]. Each goal encompasses a number of targets (169 in total), measured through the use of indicators (232 in total) [2].

Within the SDG agenda, special attention is given to inequality. It is recognized that, although income inequalities between countries have reduced, they have increased within countries and between people [1]. Inequality is associated with many critical developmental issues such as income, justice, the right of expression, accessibility to jobs, education and health, the provision of crucial infrastructures and many more. In the area of transportation, the focus of this paper, inequality is also

a key concern. The ability of people to reach opportunities, which a transportation system seeks to provide, is unequally divided in many cities, regions and countries around the globe.

Transportation-related concerns are distributed over five goals and targets: Target 3.6, related to road safety; Target 7.3, related to energy efficiency; Target 9.1 related to sustainable infrastructure; Target 11.2, related to urban access; and Target 12.C related to fossil fuel subsidies. In this paper, our prime interest is the target of urban access, which is the key target related to urban transportation. It is defined as follows: "By 2030, provide access to safe, affordable, accessible and sustainable transport systems for all, improving road safety, notably by expanding public transport, with special attention to the needs of those in vulnerable situations, women, children, persons with disabilities and older persons" [1]. Target 11.2 has as its defined indicator: "Proportion of population that has convenient access to public transport by sex, age, and persons with disabilities" [1].

Although the SDGs have been around already since 2015, little has been published in terms of a critical reflection on the choice, the relevance, and the operationalization of SDG indicators. Most publications relate to measuring progress towards the goals, for a particular country or geographical region, or a particular sector. For instance, Allen, Metternicht and Wiedmann [3] review the national implementation of the SDG-process, discuss approaches and identify gaps in 26 countries; while Vanham et al. [4] review progress on the water stress indicator. Sector-wise, most published work is found in the health sector. Maurice [5] for example discusses the science of measuring progress towards the SDGs for health, quoting Jeffrey Sachs who points out that "Indicators that are well measured, accurate, timely, and relevant for every part of the world are certainly not at hand right now, and it's going to take a number of years before they will be". Murray [6] reflects on the SDG-related health targets and proposes corresponding indicators, mainly based on considerations of data availability and scientific rigor. In a comment in *The Lancet*, Sridhar [7] discusses whether the SDGs are indeed measurable and attainable in 188 countries by drawing on the Global Burden of Disease database. She concludes that for most developed countries the indicators are relevant and measurable, but that this cannot be established for developing countries.

The UN operates a so-called tier system, which classifies each individual indicator at one out of three levels: Tier 1: Indicator is methodologically and conceptually clear, data are regularly produced for >50% of countries. Tier 2: Indicator is methodologically and conceptually clear, but data are not regularly produced. Tier 3: No methodology is yet available, still to be developed or tested. Every year, the tier classification is updated to evaluate progress towards the aim of only Tier 1 indicators. In the transport sector, the systematic collection of data at the national and local level provides a mixed picture, which is why the above-mentioned transport related indicators cover all three tiers.

To date, no work reflects on the SDG indicators in the transport sector, nor studies that reflect on the use and application of the indicators chosen, the methodological issues and the availability of data. To fill this void, we analyze its conceptual clarity, strengths and weaknesses and we reflect on the ability of the indicator to deal with inequality.

### 1.2. Shifting Paradigms in Transport Planning

In this section, we provide a brief overview of key debates in urban transport planning to position indicator 11.2 and to align the urban access indicator with those debates.

The conventional transportation planning paradigm, which originated in the 1950s in the USA, can best be described with "predict and provide" [8,9]. Its scope was to predict (car) traffic flow, on the basis of the locations of people and activities, and then provide the necessary transportation (mainly road) infrastructure to accommodate traffic. This paradigm was mobility based, and its goal was to maximize distances people can travel within their time and money budgets [8]. The worldwide application of this approach has been instrumental in the sprawling of urban regions, the reduction of urban densities and the increase in trip lengths. This predict and provide-type planning has received much criticism, and is generally acknowledged as unsustainable [10–13].

Since the 1990s, the focus shifted to the integration of land use and transportation, based on the premise that transport planning needs to be undertaken with due consideration of appropriate land use functions to reduce trip distances, and provide better options for walking, cycling and public transport. Increasingly, mobility was seen as a means to an end, rather than an end in itself, i.e., as a means to reach activities and opportunities [8,14,15]. Central to land use and transport integration is the concept of accessibility, which can be described as "the potential of opportunities for interaction" [16]. We follow Handy and Niemeier [17] who define accessibility as: "The spatial distribution of potential destinations, the ease of reaching them, and the magnitude, quality and character of the activities found there". In this definition, the degree of accessibility is determined by the location of individuals, the location and nature of the destinations and activities they engage in, and factors associated with the transport system; in most cases, it is a combination of all three. Interventions in the land use and transport planning system can, therefore, be measured using accessibility indicators. Accessibility can contribute to the integration of economic, environmental and social goals, facilitate communication between various disciplines and improve the planning process [18], which means that it is closely related to the concept of sustainability, which came up strongly in the transport sector in the 2000s and has proven its usefulness as an encompassing paradigm.

In connection to this, and in line with the discussion above on inequality related to the SDGs, attention is increasingly paid to concerns of transport (in)equality, social exclusion and (in)justice. Transport system planning and development is often carried out without explicit concern for the effects on and benefits to different groups in society, resulting in inequality. This can be seen in all kinds of contexts worldwide, in particular also in the Global South, where a lack of transport options often leads to social exclusion. In the last 10–15 years, scholars have recognized that equity concerns, which we define here loosely as the provision of a fair transport system, are at the heart of transport planning [19–24] and that proper transport planning therefore needs to consider providing benefits to all people, irrespective of their income, ethnic or religious background, housing, location, mode of travel or other factors, see Schaeffer and Schaz, 1975, cited in [9]. In trying to measure transport equity, accessibility is emerging as the central concept [22,25–27], because of its compliance with justice theory such as Rawls, see, e.g., [22,25].

This brief overview of urban transport planning debates shows that transportation planning is moving from an orientation on providing infrastructure/mobility to ensuring people's ability to reach activity locations. Hereby the focus is shifting to improved environmental sustainability through multi-modality (providing multimodal integration, promoting the use of walking, cycling and public transport and ensuring a good mix of transport modes) and social sustainability in the promotion of transport equity. It should be emphasized that the concept of accessibility is in line with the concept of sustainable transport as well as transport equality, see, e.g., [23].

Therefore, we argue that the SDG indicator 11.2 should be based on accessibility, directed at public transport and non-motorized transport and allow for the identification of inequalities. We use the term accessibility for indicators that combine both the transport and land-use system. The term access is used only with reference to the current SDG indicator, in the meaning of access to infrastructure.

Based on the above discussion, we review and implement the urban access indicator. We are particularly interested in how the defined SDG indicator 11.2 works, whether it can be appropriately operationalized, whether it provides meaningful results and is also able to capture transport inequalities. To this end, the paper addresses the following questions:

1. What, if any, are conceptual shortcomings of SDG indicator 11.2 in its current formulation?
2. How does the implementation of SDG indicator 11.2 work? How does the indicator perform in practice in a city in the Global South? Does it provide meaningful results in analyzing the state of the transport system and in explaining inequalities? Does it allow for the quantification of progress towards a better urban transport system in the context of the spatial organization of society?

3.    Can accessibility indicators be developed that are more in line with transport planning paradigms and concerns on transport inequality? How does SDG indicator 11.2 perform in comparison to these accessibility indicators?

To answer these questions, we implement SDG indicator 11.2 for the city of Bogotá, Colombia, and contrast the results with two alternative (accessibility) indicators.

The remainder of the paper is structured as follows: The next section reflects on the defined SDG indicator 11.2, vis-a-vis the previous discussion on transport planning paradigms, followed by an introduction to the city of Bogotá, a section on indicator development, methods and data used and a results section in which the results of SDG indicator 11.2 and two accessibility indicators are presented. Finally, we discuss the meaning of these results and conclude on SDG indicator 11.2 and alternatives.

*1.3. Concept and Operationalization of SDG Indicator 11.2*

The custodian agency of goal 11, UN Habitat, is responsible for the conceptual development and operationalization of the indicators under this goal. UN Habitat has developed a detailed metadata document in which the conceptual considerations for SDG indicator 11.2 are elaborated and a methodology to operationalize the indicator is provided [28]. In this section we review these considerations and reflect on some of the methodological implications of the indicator.

The UN defines SDG indicator 11.2 as "the proportion of the population that has convenient access to public transport" [1]. This raises the first question: why public transport (only)? The motivation given [29] is rather indirect, but refers to the SDG's imperative to make cities more inclusive by moving away from car-based travel to public transport and involve active modes of transport such as walking and cycling with good inter-modal connectivity. Also, reference is made to the environmental benefits of public transport. As such, the argument for public transport is in line with current paradigms, although including non-motorized transport also would be even more so. In the operationalization of SDG indicator 11.2 access to public transport is considered convenient when an officially recognized bus/public transport stop is accessible within a distance of 500 m from a reference point such as a home, school, workplace, market, etc. This distance of 500 m is a common distance used by transport planners and scientists, also because it corresponds to a walking time of approximately five minutes.

The indicator is transport supply oriented. It measures the supply of public transport infrastructure. The more infrastructure is supplied, the better the indicator scores. It does not combine demand (the actual trips people make and/or the activity locations that people travel to) with supply (the transport system and its characteristics). Transport infrastructure supply is a necessary condition for transport systems to function, but it is an insufficient condition. We consider this supply orientation as a major shortcoming of SDG indicator 11.2. A community may score very well, as they have a bus line nearby. However, if their location requires a long trip to opportunities such as work, education, health care and shops, their accessibility situation is not favorable. Paradoxically, the concepts section in the SDG metadata document [28] mentions: "Achieving SDG 11 requires a fundamental shift in the thinking on transport, with the focus on the goal of transport rather than on its means". In our view, this is precisely the problem with the definition of SDG indicator 11.2; it focuses on its means (the infrastructure), and not on the goal (accessibility). Contrary to the intention of UN Habitat, the indicator is actually mobility based, and in line with the predict-and-provide paradigm [8,9].

Another major disadvantage of a supply-based indicator in transport is that it does not address whether people are really using the supplied infrastructure (as by definition people have a number of transport options) and how people are using it. There may be many reasons, also in the context of cities in the Global South, why people would not choose a public transport system; it may be too expensive (particularly important for the urban poor), too time-consuming, too crowded, too unsafe or too uncomfortable, not available when needed, etc.

Furthermore, SDG indicator 11.2, in its operationalization [28], is exclusively directed at formal, scheduled public transport systems. In the Global South, however, most cities rely on large informal transport systems. These are, as their informal nature suggests, unregulated or partially regulated

and often operate without fixed routes, stops and timetables. The presence of these informal systems in almost all cities in the Global South is a manifestation of the fact that the formal systems supplied by city governments are not capable of meeting the transport needs of the population [30]. The UN formulation of the indicator does not consider taxis, motorcycle taxis, para-transit like three-wheelers and other informal modes. These are important modes in many contexts and their importance is only growing with upcoming ride-share initiatives. No motivation is given why such a narrow definition is chosen. As a result, the indicator captures only a fraction of public transport in, for example the African context: according to a recent World Bank report [31] the average mode share over 14 major African cities showed 6% of formal public transport and 27% of informal public transport. It is therefore obvious that the indicator does not work in such a context as the great majority of travel is not captured.

## 2. Materials and Methods

### 2.1. Transport in Bogotá

Bogotá, the capital city of Colombia, faces considerable challenges in providing mobility for its ever-expanding population of currently around eight million people. Like many other cities of its size, it is famous for its gridlock. Among the initiatives that have been taken in the last two decades to counter this problem are the widely acclaimed TransMilenio (TM) system and the full schedule and tariff-based integration of all formal public transport into this system [32]. TransMilenio is a so-called Bus Rapid Transit System, which connects most parts of the city through dedicated bus lanes. Although successful, the system has reached its limits in terms of capacity, which is why consecutive city governments have been working to construct a metro to provide more mobility options. All these public transport projects have had as specific policy objective the inclusion of the urban poor and the provision of accessibility for this group [33,34], in response to the large spatial and social transport inequalities that are present in the city [32,35]. These inequalities remain manifest despite its well-developed formal public transport system, which is why we consider Bogotá an adequate case study to review and implement the SDG indicator 11.2.

Bogotá, like other cities in Colombia, has implemented a socio-economic stratification system in the 1980s, to classify urban populations with similar socio-economic characteristics into six classes, from the poorest (1) until the richest (6) [36] (see Table 1). It is a government administration tool that is used to assess the value of real estate units, based on a combination of poverty level, availability of public services and other local characteristics. The system is applied in such a way that the wealthiest two strata (5 and 6) provide subsidies for electricity, drinking water, sanitation and waste management for the lowest strata (1, 2 and 3) [37]. This stratification system, which features in both the population statistics [37] as well as the mobility survey [38], is also important in transportation and allows us to evaluate the inequality dimension of the various indicators discussed in this paper.

**Table 1.** Population and income class per socio-economic stratum (based on [39]).

| Socio Economic Stratum (SES) | Population % | Income Class |
|:---:|:---:|:---:|
| 1 | 10.4 | Below 1 SML [1] |
| 2 | 41.3 | 1–3 SML |
| 3 | 36.0 | 3–5 SML |
| 4 | 7.8 | 5–8 SML |
| 5 | 2.6 | 8–16 SML |
| 6 | 1.9 | Above 16 SML |

[1] SML refers to Salario Minimo Mensual, or minimum monthly salary, for 2018 established at 781,242 Colombian pesos, equivalent to 221 euros (September 2018).

Table 1 shows the population distribution and income across the strata. Figure 1 below shows the spatial distribution of the various strata in Bogotá. The lower strata (1 and 2) are mostly located in the periphery of the city, particularly in the south and west, whereas the high-income strata 5 and

6 are clustering in the north. The majority of the population of Bogotá (51.7%) is classified in strata 1 and 2, which means that over half the population of the city lives in poverty, 10% of which live in severe poverty.

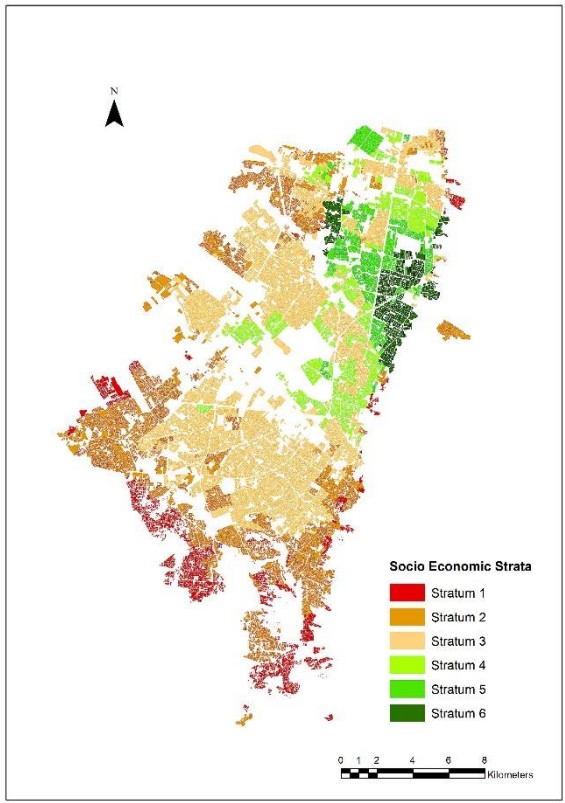

**Figure 1.** The socio-economic strata in the city of Bogotá, (based on [38]).

Income is an important determining factor in how people get around in the city. Figure 2 shows the variations of mode choice for all trips over the socio-economic strata derived from the mobility survey of 2015 [38] The figure compares data from the 2015 survey with the 2011 survey. For more detail on the survey, refer to Appendix A.2.

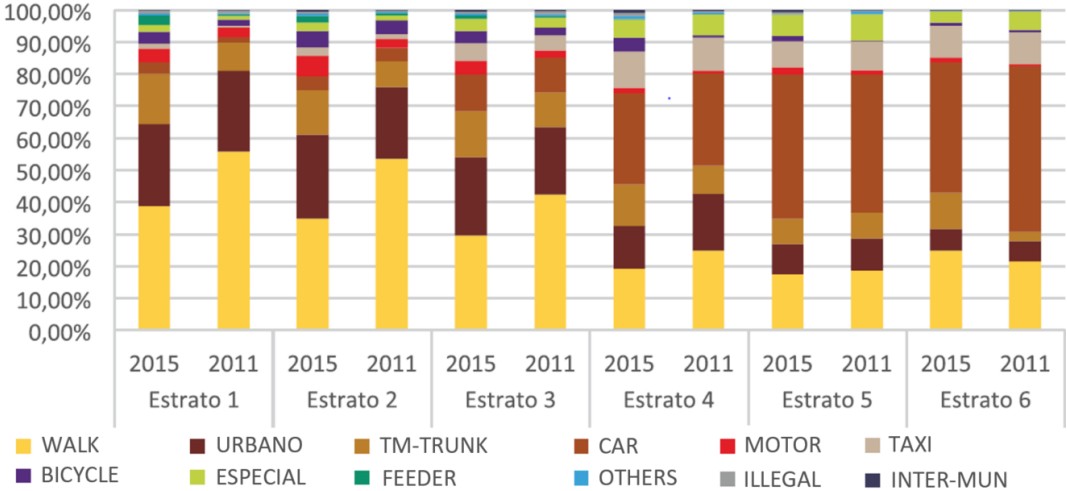

**Figure 2.** Mode distributions by socio-economic stratum (adapted from [38]). TM-Trunk, Urbano, Especial, and Feeder are all part of the integrated public transport system, Inter-Mun. refers to long distance intercity buses that are not relevant for our study. For further explanation of each system refer to Appendix A.1.

It is clear that the combined share of non-motorized and public transport for the low-income groups is very high (90% for stratum 1, 85% for stratum 2, in 2015). Contrarily, the use of private motorized vehicles (car and motor) by these strata is limited with around 8% and 11% of trips for strata 1 and 2 respectively. For the highest income groups the situation is quite different with private motorized vehicle use at 47% and 42% for strata 5 and 6, respectively. There are quite some shifts in mode share in the relatively short period of 4 years between the surveys, particularly for the lower strata. Here we observe a strong drop in the rate of walking trips, against an increase in TM-Urbano and TM-trunk trips, and cycling trips. This may be partly caused by the further physical and tariff integration of both systems [40,41] and the continuous development of the bicycle system and its integration with TransMilenio [42,43].

Below, we will discuss what bearing the inequalities in mode share and the unequal spatial distribution of residential locations have on accessibility.

### 2.2. Indicator Development

We first discuss the requirements an SDG urban accessibility indicator should meet, according to us. Based on these requirements and a brief discussion of related literature, we develop an SDG indicator based on accessibility. As discussed above, we are essentially looking for an accessibility indicator that combines supply and demand. Because an SDG indicator should have worldwide validity, and needs to be universally applied, we consider a number of criteria to be important:

1. Methodologically clear and simple;
2. Data generally and globally available or can be generated without too much difficulty;
3. Easily interpretable, unambiguous and communicable results;
4. Potential to be used in local urban and transport planning agencies.

(Note that criteria 1 and 2 connect to the concept of tiers introduced earlier, criterion 3 is a general requirement [2], criterion 4 a necessary condition to develop and implement the indicator).

There are many different conceptualizations of accessibility indicators (see, e.g., [15,17,44–46]), which creates challenges for efficient communication to policymakers and other stakeholders. Geurs and van Wee [15] distinguish four types of accessibility measures: infrastructure-based, location-based, person-based and utility-based.

Infrastructure-based measures are essentially transport system supply or performance measures. Strictly speaking we do not consider them to be accessibility measures, as they only evaluate the transport system side of the land use-transport system. SDG indicator 11.2. is an example of such a measure. As discussed above, these indicators are not in line with current transport paradigms.

Location-based measures combine the performance of the transport system with location characteristics. These measures are true accessibility measures in the sense that they are the product of both the land-use and the transport system. They are typically macro-level measures and as such useful for an SDG indicator.

Person-based measures analyze accessibility at an individual level, taking into account individual constraints such as time budgets, activity requirements and locations. Such measures require detailed individual, disaggregated data. It is difficult to analyze, aggregate, or draw lessons from such measures for an entire population. They are of interest mainly for researchers and not used in professional organizations in the global North [47]. We also consider these measures not useful for an SDG indicator.

Utility-based measures in addition are based on the analysis of economic benefits that people derive from being able to reach spatially dispersed activities. Accessibility is viewed as the result of a set of transport choices that depend on the utility of the individual's choice relative to the utility of all choices [17]. Utility measures are more disaggregated and complex than the location-based measures and require more data. Utility-based accessibility indicators can be used to measure whether interventions in the transport-land use system as a whole provide a higher utility. This is at odds with the SDG requirement of inclusiveness and inequality. It could well be that a transport system

improvement which only benefits car users for example, has the highest utility. Utilitarianism is, therefore, often discussed in the social and transport justice literature as being at odds with principles of justice [22,25,26].

With regard to criterion 4, it is interesting how practitioners in municipal planning and transport organizations—the organizations primarily involved in data collection, analysis and use— are applying accessibility indicators. Boisjoly and Al Geneidy [47] investigated the awareness of accessibility as a concept and the use of accessibility metrics in Canadian, US and European cities. Whereas 90% of the respondents indicated they were familiar with the concepts, 78% were familiar with the metrics, but only 55% used them. The metrics used were infrastructure-based or land use-based proxies and location-based measures. Of the latter, the majority used cumulative opportunity measures and the minority used gravity-based measures. None of the 343 respondents used person-based or utility-based measures. Although there is no research available on the uptake of accessibility as a concept in cities in the Global South, it is likely that reported hampering issues such as data availability and a lack of knowledge [47] will be even more pressing.

From the above, and in view of the criteria defined, we conclude that an SDG accessibility measure should be a location-based measure. Within location-based measures we can distinguish contour and gravity-based measures. An often used contour measure is the cumulative opportunity measure, which measures the potential number of activities (of given type) that can be reached from a given location, using the transport network with given modes, for a given cost (distance, time, monetary cost). Following Handy and Niemeier [17], we consider this type of measure to be a potential measure, as it measures all potential opportunities and summarizes them in a given contour travel bandwidth.

Gravity-based measures discount jobs that are further away, because nearby jobs are more attractive [15,18,48,49]. However, the development of distance decay curves in the gravity-based approach is a somewhat cumbersome and often not very transparent process, as data need to be fitted to a mathematical representation. While a gravity measure represents travel behavior better than a contour measure, the latter is simpler to generate, interpret and communicate [15,47], which is a decisive advantage for an SDG indicator. Therefore, we prefer contour-based measures and firstly choose a cumulative opportunity measure, which for the remainder of this paper will be referred to as the "potential accessibility indicator".

In addition, we propose a contour indicator that is based on actual travel and that can represent the transport reality in Bogotá well. This indicator has the advantage of being able to overcome the problem of a lack of representation of actual travel behavior and can also be developed relatively easily for different mode and trip purpose combinations. We develop this indicator in four different types, based on trip mode and purpose combinations as discussed in Section 2.3.2. In the remainder of this paper, this indicator will be referred to as the "actual travel indicator(s)".

*2.3. Methodology*

In this section, we discuss the methodology applied to indicator operationalization. The methodology developed for SDG indicator 11.2 is described in the UN metadata document [28]. We do not deviate from this methodology and, therefore, do not elaborate this further in the paper. In Appendix A.2, we describe how this indicator is operationalized in the context of Bogotá.

For the proposed indicators on potential accessibility and actual travel an elaboration is provided below. The operationalization of these indicators is also described in Appendices A.3 and A.4.

2.3.1. Potential Accessibility Indicator

In the development of a potential accessibility indicator we choose employment as the main activity at the destination, as employment trips are necessary trips for a large part of the population and are often applied in accessibility indicators (see, e.g., [18,32,41,49]). Also, in Bogotá, employment is the most reported trip purpose. As discussed, we require the indicator to be relatively simple in its formulation, easily quantifiable and easily understood. To this end, we choose a cumulative

opportunity indicator based on travel time, which is calculated on the basis of the spatial distribution of inhabitants and employment opportunities. As an indicator, we choose the ratio of the cumulative number of jobs accessible from each zone within a certain travel time over the total number of jobs available. Expressing it as a ratio corresponds to the way in which most SDG indicators are formulated and allows for easy interpretation. The indicator can be expressed generically as follows:

$$A_i^{\langle T_l, T_u \rangle} = \frac{\sum_{j=1}^{n} D_j \cdot \phi(t_{ij})}{\sum_{j=1}^{n} D_j} \qquad \forall i \tag{1}$$

with:

$$\phi(t_{ij}) = \begin{cases} 1 & \text{if } T_l \leq t_{ij} \leq T_u \\ 0 & \text{otherwise} \end{cases} \tag{2}$$

where: $A$ is the potential accessibility in location $i$, $D_j$ is the number of employment opportunities at location $j$, $T_l$ and $T_u$ are the lower and upper threshold for network travel time $t_{ij}$ and $n$ is the total number of Transport Analysis Zones (TAZs, for more details, see Appendix A.1).

For both the development of the contour-based potential accessibility indicator as well as the indicator based on actual travel, we require travel time thresholds $T_l$ and $T_u$. As we want to include all trips below a certain maximum threshold, we choose $(T_l)$ at 0 min. For $T_u$ we choose 45 min for both indicators, which we base on earlier studies and empirical data. A key concept in this connection is the so-called "law of constant travel time" developed by Hupkes [50]. In his research, Hupkes compares findings from 11 different studies and shows that the total daily travel time of people is remarkably similar, at an average of around 75 min per day.

A more recently developed concept is that of acceptable travel time [51], which departs from the idea that what people find acceptable is a combination of travel-related benefits and benefits at the activity location. Not surprisingly, Milakis et al. [51] found that the level of satisfaction of commuting trips reduces with increased travel time and varies with the mode used. For all modes on average, they found an acceptable travel time of 42.5 and 36.4 min for Berkeley, California, and Delft, the Netherlands.

The findings of Hupkes [50] and Milakis et al. [51] were mainly from cities in the Global North. It would be interesting to investigate whether large urban areas with millions of inhabitants such as Bogotá feature comparable travel times, in conditions where large sections of the population are more restrained and less flexible in travel options and destination choice. Unfortunately, to date there is no literature on acceptable travel time based on cities in the Global South. Therefore, we obtain an estimate of acceptable travel time from the Bogotá travel survey, assuming that in the situation people find themselves, concerning housing location and activity location, and given the modes that are available to them and other constraints such as budget, people do make a choice to travel a certain time. The average travel time of all 147,000 trips in the Bogotá travel survey is 49.6 min. The median is somewhat lower, 39 min, because a relatively small number of very long trips influence the average. On the basis of both the acceptable travel time discussion and the empirical data, we choose a travel time of 45 min for both indicators.

### 2.3.2. Actual Travel Indicator

We consider an indicator based on actual travel of interest for an urban transport SDG indicator, as it is a measure of how the actual transport system performs. We define the indicator as the proportion of people who access their activity location using a specific mode-purpose combination $\delta$ within a threshold of T minutes. We consider four mode-purpose combinations: (a) all public transport trips combined, (b) all public transport trips with a work purpose, (c) all public transport and non-motorized transport trips combined, and (d) all public transport and non-motorized transport trips with a work purpose. The indicator can be expressed as follows:

$$A_{i\delta}^{\langle T_l, T_u \rangle} = \frac{\sum_{r_i} \Omega\left(t_{r_{i,\delta}}\right)}{R_{i\delta}} \forall i, \delta \tag{3}$$

with:

$$\Omega\left(t_{r_{i,\delta}}\right) = \begin{cases} 1 & \text{if } T_l \leq t_{r_{i,\delta}} \leq T_u \\ 0 & \text{otherwise} \end{cases} \tag{4}$$

where: $A_{i\delta}$ is the actual reported travel accessibility within travel time thresholds $T_l$ and $T_u$ for mode-purpose combination $\delta$ in location $i$, $t_{r_{i,\delta}}$ is the reported travel time for mode-purpose combination $\delta$ for respondent $r_i$ in zone $i$.

To operationalize this indicator, we use the Bogotá travel survey of 2015. The survey is rich in content, containing both socio-economic data and travel pattern data, geocoded at the level of TAZs (for more details, see Appendix A.1), allowing for diverse types of analyses.

### 2.4. Data Used

Table 2 provides an overview of the variety of data sources used in this study; a combination of socio-economic, built environment, transport system and travel behavior data. Important aspects of these data sources, data processing and indicator operationalization are discussed in Appendix A.1.

**Table 2.** Overview of data used. * Indicator 1 refers to SDG indicator 11.2; Indicator 2, proposed by the authors, refers to the potential accessibility indicator; Indicator 3, proposed by the authors, refers to the actual travel indicator(s).

| Data | Description | Source | Indicator * |
|---|---|---|---|
| 1. Population data | Population per neighborhood based on the 2005 census | DANE, the Colombian National Statistics Office | 1, 2 |
| 2. Public Transport system | Public transport systems in operation in Bogotá, their schedules and associated geographical data. | SITP, the Bogotá integrated Transport System | 1, 2 |
| 3. General road network | The road network of Bogotá that is used for all transport modes (car, bus, walking, cycling) | Municipality of Bogotá | 1, 2 |
| 4. Building blocks | Geographic information system (GIS) data with building blocks by SES | Municipality of Bogotá | 1 |
| 5. Mobility Survey | Socio-economic data and activity pattern data of approx. 20,000 respondents | Municipality of Bogotá | 2, 3 |

## 3. Results

The following section explains the results of the three indicators. In this section and the discussion section, we focus on how well the indicators represent the transport reality in Bogotá, in terms of infrastructure provision and performance and in terms of travel demand, and how well it allows for evaluating transport inequalities and designing policy interventions.

### 3.1. SDG Indicator 11.2

The map in Figure 3a shows the result of the SDG indicator 11.2 calculation. A 500 m service area over the network around all public transport stops has been determined through a service area analysis and by dissolving all service area polygons into one single polygon. Through the overlay of the socio economic stratum (SES) residential areas, it can be seen that almost all residential areas are covered. The overall indicator score is 92%, meaning that 92% of the population has a bus stop within 500 m walking distance over the street network. This implies a nearly perfect urban access situation in Bogotá. If we look at the indicator score per socio-economic stratum (Table 3), we find some variation

between the strata, however, even the lowest scoring stratum (SES 6, the highest income population), still has a rather high coverage of 72%.

Evaluating the data differently, looking at the density of public transport stops, we observe quite some spatial variation across the city. The density varies between 1–86 stops per km$^2$ (see Figure 3b). Here we do observe inequality; in the central areas of the city where most of the commercial functions are clustered and where mostly middle-income people reside, the public transport stop densities are generally higher than in the low-income areas (see also Table 3).

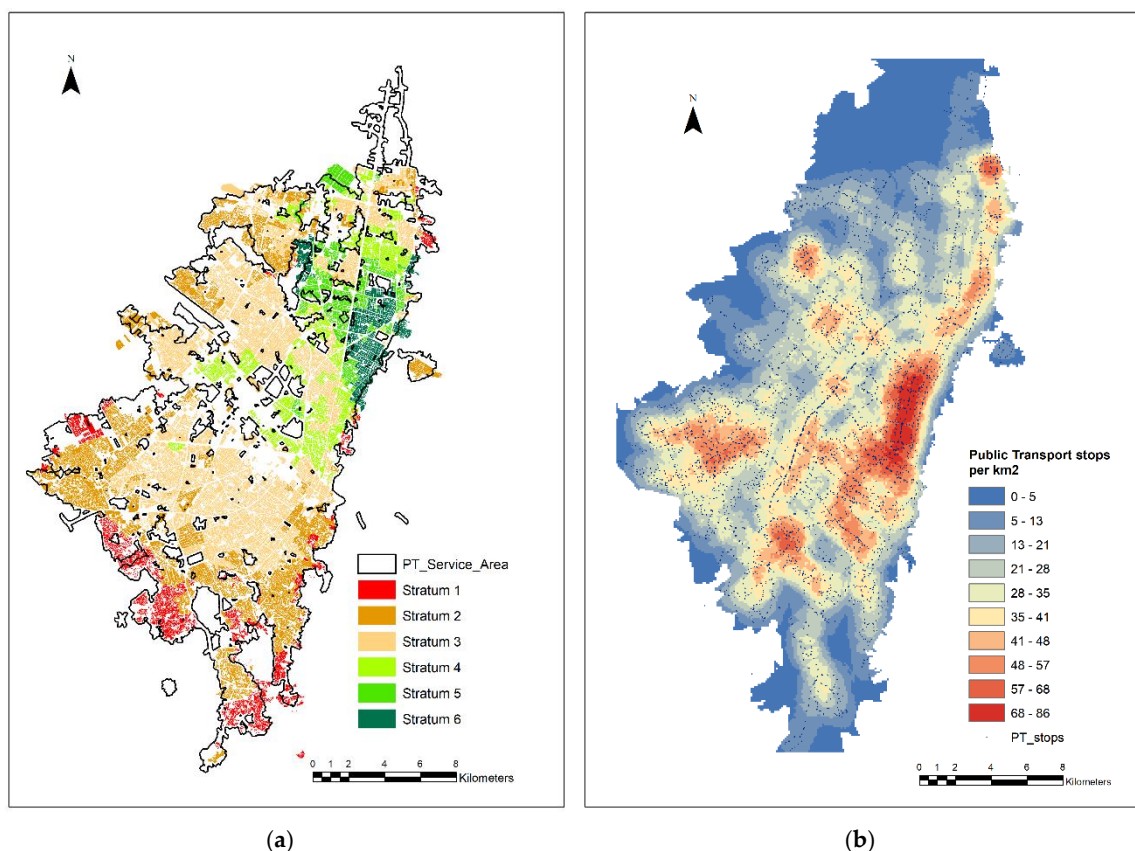

(**a**)          (**b**)

**Figure 3.** Resulting map of Sustainable Development Goals (SDG) indicator 11.2 depicting Service area of public transport (**a**), and the density of public transport stops (**b**).

These higher densities cannot only be associated with average shorter walking distances but also with a larger number of options to get onto different public transport lines and, in general, better accessibility to facilities. Because of its formulation, this inequality is not captured by SDG indicator 11.2.

**Table 3.** Quantification of SDG indicator 11.2 per socio-economic stratum.

| Socio Economic Stratum (SES) | SDG Indicator 11.2 Score (% of People Having Access) | Avg. Bus Stop Density (nr of stops/km$^2$) |
|:---:|:---:|:---:|
| 1 | 87% | 52 |
| 2 | 93% | 68 |
| 3 | 96% | 74 |
| 4 | 92% | 83 |
| 5 | 83% | 68 |
| 6 | 76% | 52 |
| Average | 92% | 70 |

### 3.2. Potential Accessibility Indicator

To derive the potential accessibility indicator, an origin-destination matrix of 849 origins and 790 destinations (only those zones where jobs are located) was derived, resulting in over 600,000 potential trips. Out of these trips, 40% were associated with journeys under 45 min by public transport. Figure 4 shows the resulting accessibility values and distribution over the city.

We note a concentration of employment in the central eastern part of the city. As a result, the accessibility to all jobs that are within 45 min travel time is also the highest in this area. In the green zone, 60–80% of all jobs in the city are accessible within 45 min by public transport. Despite the concentration of jobs in the central eastern part, there is no location in the city from which more than 80% of all jobs are accessible within 45 min. This is due to the presence of jobs in peripheral areas, although concentrations are low. The result of the potential accessibility indicator displays an almost perfect concentric model, which can be explained by the pattern of jobs in combination with the travel time on a very dense public transport network. All areas of the city are well connected to public transport, as we observed already with SDG indicator 11.2 (full supply coverage), and the 400 routes cause a flat public transport landscape, where travel time becomes a function of distance. We cannot observe an effect of the TransMilenio bus rapid transit (BRT) system. Although the speed of that system is somewhat higher than the other bus systems, this effect is too little to observe in the maps of the potential accessibility indicator (Figure 4b).

With respect to this indicator, there is a clear equality issue. The areas in the south, where most of the lowest income group reside, can reach less than 20% of all jobs in 45 min, whereas the more central locations with higher incomes score much higher.

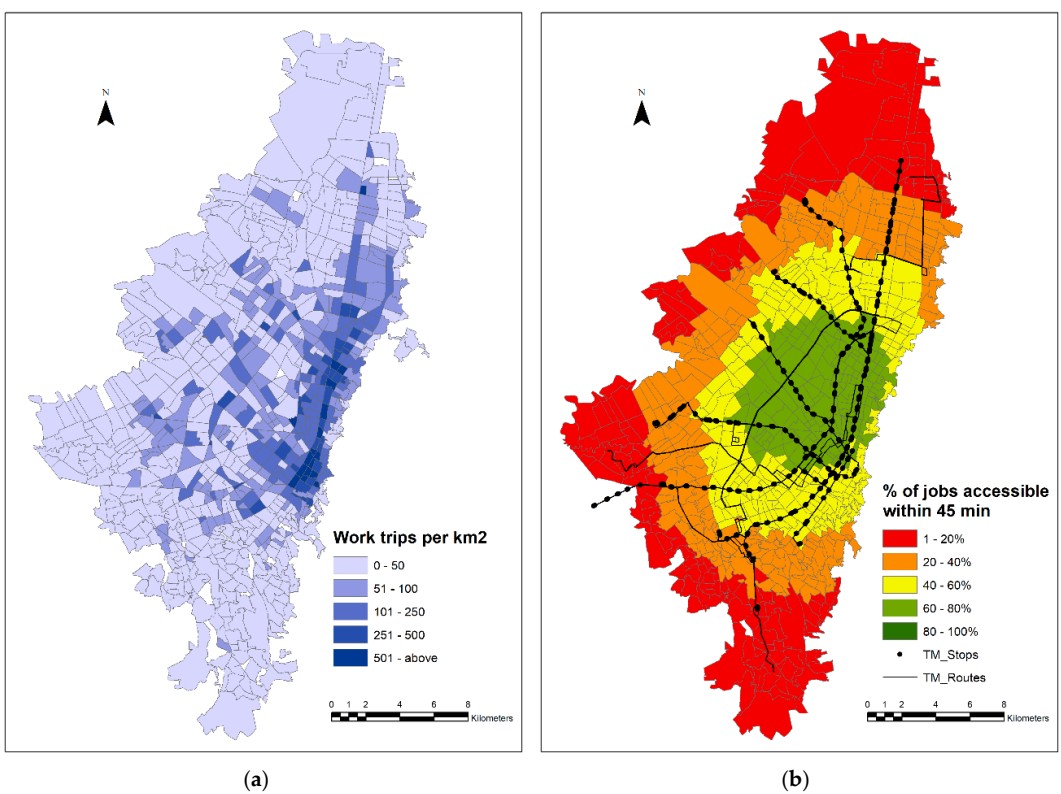

**(a)**        **(b)**

**Figure 4.** The number of job destinations of survey respondents by zone per km$^2$ (**a**) and the % of jobs that is accessible within 45 min from each Transport Analysis Zone (TAZ) (**b**).

### 3.3. Actual Travel Indicators

In this section, we develop four indicators based on actual travel. For each of the 849 zones, the average travel time of all home-based trips has been determined and the proportion of trips that

is lower than 45 min has been calculated and assigned to the zone. We first show the results for two public transport based indicators (all trips, work related trips), followed by two indicators for public transport and non-motorized transport (walking and cycling; all trips, work-related trips). The maps in Figure 5 below show the results of the actual travel indicators.

Contrary to what we observed earlier with SDG indicator 11.2, we see here that the public transport accessibility in Bogotá, based on revealed travel data, is not favorable at all. In the biggest part of the city, less than 40% of the public transport trips made are shorter than 45 min. This is even worse for work-purpose trips made with public transport. Large sections of the city score below 20%. We also observe that the spatial distribution of accessibility is not uniform. Towards the fringes of the city, travel times are becoming longer and, therefore, the percentage of people who reach their destination within 45 min reduces. This is clearly a situation of spatial inequality. If we compare the accessibility across socio-economic strata, we also see quite some variation. Table 4 below summarizes the averages for the different strata.

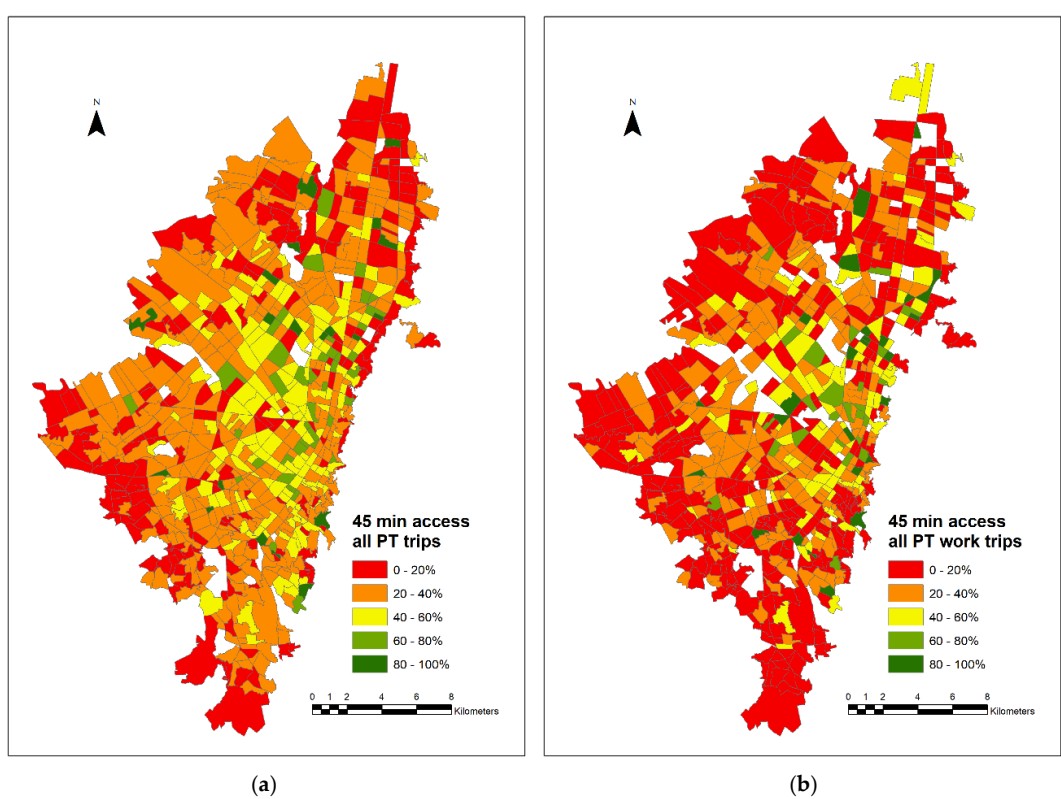

(**a**)                    (**b**)

**Figure 5.** The proportion of people who reach their activity location within 45 min by public transport (PT), for all trip purposes (**a**) and for work trips (**b**).

The combination of public transport and non-motorized transport trips (Figure 6 below) shows much better accessibility values, particularly in 5a; only few areas score lower than 40%. This is due to the fact that walking and cycling trips have been included, which brings in the shorter duration trips. For this indicator, the majority of people are in the middle category, where 40–60% reaches their location within 45 min. Including walking and cycling also provides a picture of reduced spatial inequality, the differences between the more central locations and the fringes are reducing. This can be clearly observed if we compare Figure 6a to Figure 5a, but also if we compare it with 5b and 6b. The better performance in terms of spatial inequality is also confirmed through the comparison of stratum equity values, as reported below in Table 4. Also here we see that the performance of the transport system for work trips is much worse (Figure 6b). However, it is better than the results in Figure 5b, which implies that there is a considerable number of people who reach their work location with non-motorized transport also.

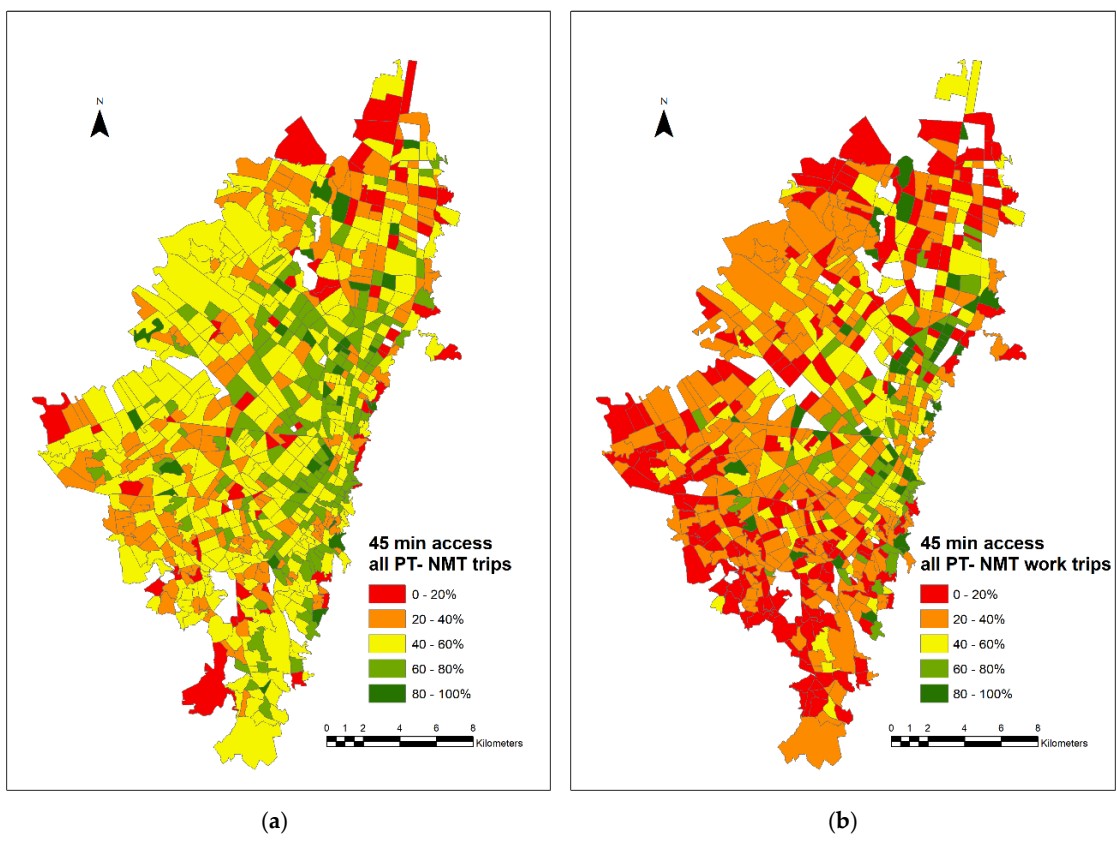

(**a**)                                                (**b**)

**Figure 6.** The proportion of people who reach their activity location within 45 min by public transport (PT), walking or cycling (NMT), for all trip purposes (**a**) and for work trips (**b**).

The effect of travel time and purposes can be seen in Figure 7, which gives trip purposes with the associated number of home-based trips and average travel time. The top three categories are medical trips, work trips and administrative trips, all being activities clustered in the more central locations in the city far away from the periphery.

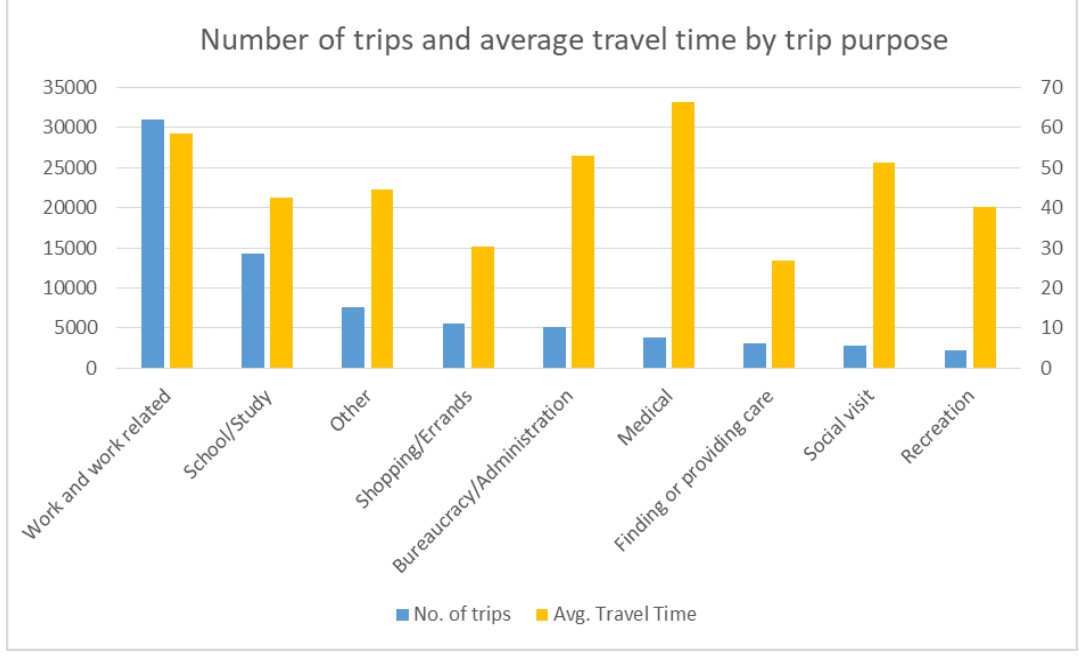

**Figure 7.** The number of trips and average travel time by trip purpose for all home-based trips.

The trips that have much shorter travel times are to shopping and care locations; these are closer and these trips are therefore also more often done by non-motorized modes. For the city as a whole, for all trips in the database which have their origin at the residential location, i.e., home-based trips (76,312 trips), 57% of all people reach their destination within 45 min, leading to a total daily travel time of 1.5 h.

**Table 4.** Quantification of all indicators per socio-economic stratum.

| Socio Economic Stratum (SES) | SDG Indicator 11.2 | Potential Accessibility PT-Work Trips | Actual Travel Indicator PT Trips | Actual Travel Indicator PT-Work Trips | Actual Travel Indicator PT&NMT Trips | Actual Travel Indicator PT&NMT Work Trips |
|---|---|---|---|---|---|---|
| 1 | 87% | 16% | 20% | 15% | 42% | 19% |
| 2 | 93% | 26% | 28% | 20% | 47% | 29% |
| 3 | 96% | 46% | 33% | 27% | 50% | 36% |
| 4 | 92% | 57% | 38% | 33% | 50% | 40% |
| 5 | 83% | 47% | 34% | 30% | 46% | 39% |
| 6 | 76% | 50% | 36% | 29% | 54% | 50% |
| Average | 87.8% | 40% | 31% | 26% | 48% | 36% |
| St. Dev. | 6.8% | 14.2% | 5.9% | 6.4% | 3.8% | 9.7% |

Table 4 provides an overall comparison of stratum-based inequalities as measured with all indicators. In contrast to SDG indicator 11.2, achieving very high scores, all other indicators have much lower transport accessibility scores with averages well under 50%. In the case of the actual travel based indicator for public transport trips to work, only a quarter of the people reach their work location within 45 min of travel. This is undoubtedly the worst scoring indicator. People in the lowest strata, who are generally further away from job opportunities (see also Figures 1 and 4a), score values of 15% (stratum 1) and 20% (stratum 2) respectively, as a result of their long commutes and often complicated travel pattern with one or more feeder buses and many transfers. Despite the well-developed and dense public transport system in Bogotá, we observe that public transport-based actual travel indicators score very low, owing to the fact that distances are high and average speeds are modest.

Interestingly, Stratum 4 provides the best results for the actual travel public transport indicators (albeit marginally) and certainly for the potential accessibility indicator. In the latter, given that we use the travel time to all employment locations, the central location of stratum 4 areas pays dividend, also for public transport trips.

The actual travel indicator for public transport and walking and cycling trips flattens the inequalities considerably due to the inclusion of shorter trips. People in the lowest strata have options to use these modes to reach locations closer by, mainly for non-work purposes. A lot of the shopping and social trips can be done locally and require less time. More than public transport, walking and cycling can, therefore, be considered equalizers.

Comparing the potential accessibility and the actual travel public transport non-motorized transport shows that stratum 1 and 2 are operating above their potential (in particular for all-trips), which implies that people are able to find closer work locations than corresponds to their potential. This is clearly not the case if we look at public transport trips only.

## 4. Discussion

Through the review and implementation, we find that SDG indicator 11.2. is conceptually weak and generates a picture of an almost ideal transport situation in Bogotá. The almost universal coverage of public transport stop access within 500 m only captures the transport supply side, while there is obviously variation in activities across socio-economic groups and areas and, therefore, variation in demand. The results indicate that the lowest scoring areas are the areas where the highest income groups reside. Paradoxically, therefore, to ensure that the overall indicator score goes up, investment in new public transport stops would need to be done in high-income areas. This is not a good idea from a transport equity point of view, as the residents in these areas already have more travel options

and are advantaged in terms of their accessibility, as the potential accessibility indicator shows. Also, the public transport stop density is lower in these areas for a reason, namely that there is less demand for public transport trips, which is also caused by high car ownership and use.

Another downside to SDG indicator 11.2 is that, in order to improve the indicator score by increasing the number of bus stops, the number of stops per route will increase, causing average speeds to drop, which may result in slower service and less accessibility to activities. Countering this with limited stop services could mean longer waiting times and higher operational costs. It is doubtful, therefore, whether constructing more bus stops per unit of area, to reach the theoretical maximum of 100% coverage, will improve the transport situation; it may well aggravate it. Thus, the indicator does not help to target transport system improvements across socio-economic groups and areas.

The potential accessibility indicator demonstrates that, due to the concentration of jobs in the center of the city, people living at the fringes are at a disadvantage. The indicators based on actual travel behavior show that many people make long trips and that there is significant inequality in terms of travel times, particularly for public transport work trips. People from the south, east and far north of the city, who are mostly in the lower income categories, have much longer travel times to reach their work, which corresponds to the findings of Oviedo [32].

One could expect that the actual travel indicator for public transport work-related trips and the potential accessibility indicator based on public transport trips to employment locations would provide similar results. This is not entirely the case. Although there are similarities in the order of the strata scores (strata 1,2,3,6,5,4 and 1,2,3,5,6,4 respectively, from lowest to highest), the actual travel indicator scores considerably lower than the potential accessibility indicator, particularly for the middle and higher strata. This could be due to the fact that the performance of the public transport system is in reality lower than is modeled, but it may also point to a spatial mismatch between residential location and job locations. The fact that the socio-economic strata are very clustered, is indicative of a lack of affordable housing in central locations for the lowest strata and the choice to live in a high-income area for the highest strata. Employment of different type is somewhat more spread over the city, which may explain the variation between potential and actual values. Interventions to improve the actual accessibility could be done on the transport side, through improvements in the public transport system (an example is the planned construction of the metro) and on the land-use side, e.g., through a better mixing of income groups, ensuring that low income groups also have options to find a house in a central locations. An example of a recent project is the Plaza de la Hoja social housing project; however, many more of these types of project are needed. Interestingly, the potential measure corresponds better with the actual travel public transport—non motorized transport to work measure. Had we included non-motorized transport into the potential measure, its values would probably have been very similar, as there will be few job locations in the city that can be reached in 45 min with walking and cycling that cannot be reached with public transport. Including non- motorized transport in the potential measure may therefore be useful.

The results on the actual travel-based indicator which includes public transport and non-motorized transport demonstrate its ability to adequately measure transport system performance, in particular for the lower economic strata, who are the key target group of transport policies in Bogotá. This indicator ticks all the boxes in terms of current paradigms in transport and is able to give a more balanced picture of spatial inequalities. It also shows that reality is less severe for the lower-income groups when it comes to being able to reach activity locations other than employment locations. Moreover, it can involve all public transport modes, also the informal. It may be a politically more attractive indicator then the potential accessibility indicator.

Although the potential and actual travel accessibility metrics tell a similar story, the differences are also striking and illustrative of how careful one needs to be in defining, choosing and interpreting a metric. The metric based on actual data is attractive, as it is a reflection of how the current transport system is being used by all people, across socio-economic groups. The potential metric, however, is more universally applied and attractive from an equality or opportunity point of view, see,

e.g., [19,22], in the sense that it measures how many activity locations can be reached in a given time span with public transport, which in Bogotá is the most available option besides walking. This provides for a more fair comparison between socio-economic groups.

Naturally, all indicators require data to be collected, and the newly proposed urban accessibility indicators are no exception.

SDG indicator 11.2 relies on data of public transport bus stops, the digital road network and population data at a sufficient level of aggregation. As discussed earlier, informal transport supply is not included in the operationalization. Even in its current formulation ("the proportion of the population that has convenient access to public transport"), and choosing a supply focus like the UN has done, it is hard to ignore informal transport, as it is much bigger than formal transport in most Global South contexts.

Collecting data on the (mostly informal) transport systems in the cities of the Global South may be cumbersome and time-consuming. To date, about 1000 public transport organizations have shared their data via the General Transit Feed Specification (GTFS), but this includes only a few public transport organizations in the Global South. This is a problem because in particular in this context, transport systems are highly dynamic and transport inequalities more profound. Notwithstanding the above, there are currently a lot of initiatives ongoing by tech companies to fill this gap. An example of such initiative is *WhereIsMyTransport*, who have generated data of informal public transport systems in a number of African and Asian cities already and are expanding their coverage. It is likely that in a few years' time almost all major cities will be covered by these types of initiative. Challenges will be most pressing for medium sized and smaller cities that have less resources.

For the potential accessibility indicator, the prime data sources required are the public transport system (in particular the routes, stops, headways and speeds) and information on employment or other activities (if required). The latter can usually be obtained via census data or via a household travel survey.

The actual travel indicator relies entirely on a household travel survey. Although this takes time and is expensive, it is an indispensable instrument in strategic transport planning, as it provides the basis for decisions on demand for and appraisal of new transportation infrastructure. Most major cities in the Global South have already carried out such surveys, and many are doing this on a regular basis to inform planning and decision making around infrastructure projects such as bus rapid transit. Where such data is available, the actual travel indicators can be developed relatively quickly. The practice of applying TAZs is widely embedded, which allows for a more sound comparison of outputs between cities, adding to its suitability as an SDG indicator. Generally speaking all indicators can be well developed for major cities, less so for smaller cities. On the basis of the above, we would classify all three indicators as Tier 2 at the moment.

In terms of transferability of the approach, the 45 min travel value looks reasonable for large urban agglomerations, however, more comparative research is needed into travel time distributions and acceptable travel time, particularly in the Global South to contextualize the indicators. In principle the indicators as formulated in Equations (1)–(4) allow for choosing context specific travel time thresholds. The findings from Bogotá are not necessarily transferable to other cities, which have different spatial and socio-economic arrangements. The aggregation of indicator values against socio-economic strata as shown in Table 4 might give more mixed results in other contexts, e.g., in some of the major Indian cities like Ahmedabad, where large pockets of low income population can be found in and around city centers, close to employment [49].

Table 5 provides a summary overview of the three indicators and some key characteristics. The first four key characteristics deal with how well each indicator represents transport reality, followed by its conduciveness to analyze spatial inequality, its use in evaluating planning interventions, and lastly, its SDG tier categorization.

**Table 5.** Overview of indicators against key characteristic (* = authors categorization, methodology established, data not regularly produced).

| Key Characteristic | SDG Indicator 11.2 | Potential Accessibility Indicator | Actual Travel Indicator |
|---|---|---|---|
| Captures infrastructure supply | Yes, only public transport stops | Yes, full public transport and pedestrian system | Yes, full public transport and road system |
| Captures actual travel demand | No | No | Yes |
| Captures potential travel demand | No | Yes | No |
| Captures PT performance | No | Yes | Yes |
| Allows for analysis of spatial inequality | Limited, results may have little meaning | Substantial, in terms of potential travel and equality of opportunities | More substantial, in terms of revealed travel |
| Allows for evaluation of spatially targeted policy interventions | Only in areas where there are few bus stops, but meaningfulness cannot be established | In both the public transport system and the land use system | In both the public transport system and the land use system, but effects may not be attributable to the intervention |
| SDG Tier * | 2 | 2 | 2 |

## 5. Conclusions

We conclude that the UN formulated SDG indicator 11.2 has conceptual weaknesses. Being supply-centered, i.e., on physical public transport infrastructure, it connects to the wrong paradigm. It measures one of the "means" to get to a better transport system, but not the "end" of what the transport system should deliver: whether people are able to reach their activity locations. It therefore also does not provide meaningful information for targeting improvements. In our view the provision of public transport infrastructure is a necessary but insufficient condition and measuring only this is too limited.

The results of the application of SDG indicator 11.2 to the reality of Bogotá shows that it does not provide meaningful and useful results. We expect this to be similar in most cities in the global South, as well as cities in the global North, but for different reasons.

Our analysis shows that relatively simple accessibility metrics provide a more realistic, richer and more diverse picture of how well urban transport serves people. These metrics perform better in the analysis and visualization of spatial inequalities, which paves the way for more targeted interventions in the integrated land-use planning and transport system, in line with upcoming transport justice and inequality frameworks [19,22]. In particular, the potential measure is suited for the evaluation of relative improvements in accessibility before and after interventions.

We recommend UN Habitat, the custodian agency for SDG target 11, to reframe their urban access target and indicator and replace it with an accessibility indicator that also captures demand, preferably a potential accessibility indicator of public transport or an actual travel indicator of public and non-motorized transport. We encourage cities to discuss these and other indicators in the development of locally relevant indicators that are of real use to inform policy.

**Author Contributions:** Conceptualization, M.B.; Data curation, M.B.; Formal analysis, M.B.; Methodology, M.B., M.Z.; Supervision, K.P., M.v.M.; Validation, M.B., M.Z.; Writing—original draft, M.B.; Writing—review and editing, M.Z., K.P. and M.v.M.

**Funding:** This research received no external funding.

**Conflicts of Interest:** The authors declare no conflict of interest.

## Appendix A

*Appendix A.1 Explanation of Data Used, Data Processing and Indicator Operationalisation*

Data sources used are discussed in the sequence as presented in Table 2:

1.  Population data. The population data that is available is based on the census, which was held in 2005. Early 2018 a new census was held. However, this data was still unavailable at the time

of writing. The provided data is aggregated by neighborhood, or "barrio", which is typically an area of around 50,000–100,000 inhabitants.

2. Public Transport. The city's formal public transport systems are integrated into one system called SITP, (Sistema Integrado de Transporte Público de Bogotá). The system consists of the following sub-systems:

   - TransMilenio (TM), the main system for bus rapid transit, consisting of dedicated and separate infrastructure: 12 lines, 149 stations.
   - TM Alimentador, a system that is meant as a feeder bus system that connects primarily the outer areas to the TM stations, where people can access the TM proper: 112 lines. Operates on the regular road network, also partly on unpaved roads.
   - TM Urbano: 234 lines. The major regular bus system of Bogotá that has been brought under the TM umbrella operates on normal and major streets and covers almost the whole city. Connects zones directly, not necessarily via the TM main system.
   - TM Complementario: 25 lines. Connects selected single zones directly to a TM station area.
   - TM Especial: 15 lines. Connects a few peripheral areas of the city to the TM main and TM Urbano Systems.

   The data derived for the public transport systems and roads of Bogotá was collected via the city of Bogotá and SITP. All systems operate on the general road network except the Transmilenio BRT, which has its dedicated infrastructure. We use the GTFS for the acquisition of static data [52]. GTFS is based on an agreed format for public transport systems, their schedules and geographical data. Public transport operators make GTFS data available for developers to write scheduling applications, such as used by google maps and others. This data is suitable for the quantification of SDG indicator 11.2, as all required spatial and non-spatial characteristics of the public transport system are available and this is the most problematic category of data in the quantification of this indicator as it is currently defined. For an overview of the GTFS data structure, see [52].

   This data has been converted to the GEOJSON format, imported into the ArcGIS software, where for each of the above mentioned public transport systems in Bogotá a feature dataset of the routes and the stops was created.

3. General road network. The data of the general road network and building blocks have been obtained from the municipality of Bogotá during research that took place in the city in 2015 and was further updated in 2017. It contains road center lines which have been used in the construction of the public transport network, along with the dedicated infrastructure of TransMilenio.

4. Building block data. These have been obtained from the municipality of Bogotá. Population data have been linked to these building blocks at the neighborhood level, and later disaggregated.

5. Mobility Survey. The mobility survey of 2015 [38] was carried out under auspices of the Mobility Secretariat of the Municipality of Bogotá. It is the fourth survey of its kind, earlier surveys were held in 1995, 2005 and 2011. The data consists of socio-economic data of the respondents and trip-related information such as mode of transport, duration, cost etc. Over 147,000 trips are thus recorded, based on a sample of approx. 28,000 households with 92,000 inhabitants. This leads to an average number of trips of 1.6 per inhabitant for the sample. The data is structured in so-called Transport Analysis Zones (TAZs), which is a spatial-administrative construct that is often used in transport planning [53]. The whole study area contains about 900 TAZs. We have eliminated records that show no trip duration (same start and end time of the trip), trips that were not connected to a TAZ, and TAZs where no home-based trips were recorded. Also, records with coding errors have been omitted. For some operations, we included thresholds, e.g., only those TAZs where at least 10 trips have been recorded. This is further explained below. The full data of the Bogotá mobility survey is available via the open data portal of the Colombian government.

*Appendix A.2 Operationalising the Indicators in Bogotá*

*SDG Indicator 11.2*

In first instance, we have implemented SDG indicator 11.2 for the city of Bogotá, Colombia, following the UN based operationalization [28], which lays out the following methodological steps:

1. Delineating the built-up area, or area under study;
2. Making an inventory of public transport stops;
3. Estimating the urban area with access to these stops (essentially the area that is within 500 m of any bus stop);
4. Overlaying the service area obtained in step 3 with the population data layer:

   Step 1: The Capital District of Bogotá is divided into 20 localities, 19 of which are urban, but not all of them are entirely built up. We base the built-up area on the area in which residential development is present.

   Step 2: All bus stops of the earlier defined systems (TM, TM alimentador, TM urbano, TM complementario and TM especial) have been joined together in one GIS layer to create a complete public transport system.

   Step 3: The metadata document discusses the use of Euclidian or network-based distance measures and concludes that network-based measures are preferable, as they provide a more realistic approximation of reality. Although network-based is not prescribed, we implement the measure based on network distance. To this end, we use the general road network, used by pedestrians to get to the bus stops, and the bus stops, to create service areas for each bus stop, based on the distance of 500 m. By combining all individual polygons, we create a service area polygon for the entire system.

   Step 4: In order to estimate the population having access to public transport, we use the official population data from DANE, based on the 2005 census, which is made available at the level of neighborhoods ("Barrios"). As the analysis requires a finer spatial resolution than neighborhoods, to be able to overlay the population data and the catchment polygon produced in step 3, we disaggregate the population data to the neighborhood building block boundaries per SES, in proportion to their area. This also allows us to calculate the SDG indicator 11.2 score per SES.

*Appendix A.3 Potential Accessibility Indicator*

The following steps are applied in the analysis:

1. The development of a multi-modal network in ArcGIS, which involves all types of buses. See [49,54] for more details on the methodology. The network includes all public transport stops and all routes of all TM systems, as discussed above. We base this on an average bus frequency of 10 min translating to an average waiting time of 5 min, an average bus speed of 13 km/h and an average TM speed of 25 km/h [55].
2. The determination of all zones from which work-related trips originate. From all origin zones, the centroids are determined.
3. Identification of all zones which are trip ends of a work-related trip. In total, we have some 31,000 trips that are work-related. Also from these zones centroids are constructed, containing the number of work locations per zone.
4. Construction of an O-D matrix between all origin zones and all destination zones with work activities. The matrix makes use of a shortest path algorithm over the network, and chooses the most optimal route in terms of travel time, by public transport.
5. Selection of all O-D pairs that are less or equal than 45 min, with their associated TAZs. These are linked to the table with work locations. A Pivot table is constructed in which for each origin

location the number of employment locations at the destination TAZ is summed. This table is used to create an accessibility map.

*Appendix A.4 Indicators Based on Actual Travel*

In the analysis the following steps were taken:

1. The travel survey data were imported into MS Excel. About 1% of records with missing data values or errors in either travel time duration or TAZ code were removed from the sample.
2. Four different selections of the data were made, corresponding to the four different indicators: for all public transport trips, for all public transport trips with a work purpose, for all public transport and non-motorized transport trips, and for all public transport and non-motorized transport trips with a work purpose.
3. For each of those selections, pivot tables were made of the valid records, in which for each origin TAZ, the number of trips originating from that zone was calculated. These were then classified into 6 classes based on the travel time of the trip (travel time $\leq$ 30 min, $\leq$45 min, $\leq$60 min, $\leq$90 min, $\leq$120 min, >120 min).
4. The pivot tables were imported into ArcGIS and joined with the TAZ layer that had been aligned with the study area. This operation has eliminated the outside TAZs of the more rural districts.
5. Four maps were generated with classified % scores of respondents reaching their activity location within 45 min.
6. Centroid locations of the building blocks were extracted to derive the building locations per stratum. A spatial overlay was made with the indicator maps to calculate the scores per stratum.

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
