# Peer review of "Access or Accessibility? A Critique of the Urban Transport SDG Indicator"

_ijgi, doi:10.3390/ijgi8020067_

Round 1

Reviewer 1 Report

Dear Authors,

thanks for your paper. I don't have relevant comments on this occasion apart suggesting to add more academic literature about currently available methods to quantify several types of accessibility indicators/parameters such as psycho-economical distances and many other concepts of accessibility. 

Sincerely

Reviewer 2 Report

In this paper, the authors propose two alternative measures to compare against Indicator 11.2, which is an indicator that can be found in the UN Sustainable Development goals.  The authors find that the SDG indicator – as written – has critical shortcomings that perhaps provide a falsely-inflated view of transportation access in Bogota, Colombia because of its restricted focus on transportation supply. They conduct GIS analysis, using population data, transportation network and transit stop data, and a mobility survey to operationalize the SDG indicator and the two proposed alternative indicators to assess their performance across the city. This kind of work is important to contribute to the literature at present, especially in the rapidly-developing era of Smart City efforts that include transportation-related outcomes (very much from a supply side view), and they are correct to note that the supply-side focus of the indicator is problematic and that alternate approaches are needed.  While they make some strides to address this need, there are some critical issues with the paper as written that should be expanded, addresses, or clarified before it will be suitable for publication.  

Primary General Comments:

1) The authors should formally specify the indicators that they have constructed.  They list the steps used to construct them in the Appendix, but formal mathematical specification would be better in terms of general use for other studies and applications.  These specifications would be good to place in the Materials and Methods section where each indicator is introduced.

2) Indicator 1 and the SDG Indicator 11.2 are very clearly about public transportation access.  That does not appear to the case, though, with Indicator 2, which makes comparisons between it and the aforementioned indicators difficult.  It seems like it ought to focus explicitly on public transportation use instead of being mode-agnostic, or at least highlight the degree to which observed public transportation use contributes to the observed data and spatial patterns.  This indicator relies on a 45-minute travel threshold, but the authors note it applies to any trip completed, whether through public transit or not.  That conflates an awful lot of activity spaces of different sizes using this time threshold.  They do acknowledge this limitation in the third paragraph of the Discussion, but it is unclear why they did not extract those trips completed by public transit for this Indicator, given that it seems they have the data from the mobility survey to do so. Indeed, they note in the results section of Indicator 3 that they know how many trips are associated with journeys under 45 min by public transport (Page 11, Lines 404-405).  If these data on trips made by public transit are available, why not use them, especially since the SDG Indicator 11.2 specifically addresses public transportation access?  Alternatively, if the authors truly would like to consider all modes for Indicator 2 that relies on observed travel data, could they create a new section in the results that shows how this indicator would look using only observed public transportation travel data as a bit of a comparison piece to Section 3.2?

More generally, it might be interesting to compare how indicators perform across particular modes from the mobility dataset.  Figure 3 appears to include travel by all modes, but it would be helpful to see a similar set of figures for public transportation. 

3) It might also be worth conducting some sensitivity analysis on the 45-minute threshold chosen for Indicator 2.  The authors could run this analysis using the median value found in the survey (listed on Page 8, line 313), or perhaps 30 or 60-minute thresholds which would represent more extreme values.    

4) I appreciate that the authors used network service areas instead of Euclidean distance buffers to identify coverage within 500m of public transportation stops – that is a better geographic representation of access.

5) Related to point 2, it would be worth clarifying the type of trips made by public transit.  Appendix A, Point #2 indicates that there are five sub-systems that are part of the city’s public transportation system, all of which appear to serve different trip purposes, offer different service areas/speeds/frequencies, and may be accessed differently in different parts of the city.  The authors should consider these differences in the indicator or explain more clearly why it is suitable to treat them equally for the purposes of this indicator.

6) The Introduction section should focus less on background information about transportation planning and more on indicator construction and assessment, given the stated aim of the paper, and the methods and results that follow.  Section 1.2 contains some useful and interesting information, but is quite high level in nature and could be summarized in a far more concise manner and referenced accordingly.  Instead, I would like to see the authors create a new section that addresses accessibility indicator construction, and situate their accessibility indicators in the literature accordingly.  This needs to be addressed, given what the authors aim to accomplish with this paper.  There is a large amount of literature on this topic – some of which is addressed on Page 8, Lines 330-331, but more work on this is required.

7) In the Discussion section, can the authors include some discussion about the limitations of how these findings in Bogota may not be transferable to other cities? 

Other Comments:

1) Can the authors provide some summary statistics about travel in the City of Bogota in Section 2.1, or in a different section?  It would be helpful to the reader to understand the degree to which people travel by car/public transportation/active transportation in the city, perhaps even broken down by SES zone.  That would help contextualize the findings a bit.  This may also be useful in the results section 

2) Page 2, Lines 73-75. The authors reference Tiers, which are relevant to SDG indicators.  Can the authors clarify what these Tiers mean more to a reader who may not be familiar with them?

3) The abstract can be made far more concise.  Please streamline.

4) The authors should consider tables similar to Table 3 for Sections 3.2 and 3.3 in the results to show how accessibility varies across the SES strata in the city.

5) Table 4 might be better suited for the Methods section so that the reader has an idea of what data are required to form these indicators.

6) Page 14, Lines 520-521 in the Conclusion.  The last sentence seems like it was left in the manuscript in error, or is a typo.

Reviewer 3 Report

This paper reports on the use of transport indicators to attain the Sustainable Development Goals defined by the UN. The issue is very interesting and fully falls within the scope of IJGI. Few publications have addressed the validity of the UN indicators scientifically. Much of this paper is devoted to the study of Bogota. Although the databases used remain “in the background” –few information is provided about their specific characteristics and design-, I appreciated much of the processing strategy used by the authors, i.e. the confrontation between the UN indicator and “homemade” alternative calculations. GIS structure and processing seem to be operational and consistent.

My main concern is about the purpose of this paper, and specifically the first part of the article. The authors focus their attention on one among 232 indicators. Even if their criticisms were to be listened by the UN authorities (I highly doubt that), this wouldn’t change one iota the general statistical balance. There is an obvious ambiguity here. And the conclusion expresses some kind of ingenuity.

Part 1 is the less convincing section of the paper. The issue is addressed through some kind of applied transportation geopolitics. Here the authors often push at open doors. The purpose of the UN through the establishment of SDG is not really explained in terms of objectives, legal framework, and applicability throughout the world (territorial dimension). There is a large gap between the not fully mastered conceptual approach and the example of Bogota.

Another point that leaves me rather confused is scale discrepancy. The chosen indicator is supposed to be valid worldwide but the approach exclusively focuses on urban areas, as if rural lands had vanished from the earth surface. Among these urban areas, Bogota is an excellent example, although one among … hundreds of such large urban units. Obviously, statistical and geographical representativeness are not viewed as a serious problem.

So, in general, I have mixed feelings. On the one hand the Bogota case study is addressed cautiously and validates the idea that much more efficient indicators need to be produced. On the other hand, I’m not sure the UN SDGs are the appropriate framework or pretext to address the issue.

Other tips to be considered:

1-In the first part, access and accessibility are not considered clearly according to time and distance. Later on, these two parameters are introduced, but through the back door. That’s unfortunate in my view.

2-Bogota as a case study. Why this choice among many others?

3-The relationship between HDI and SDGs is not explained. Missing background information.

4-As I see it, access and accessibility are defined in socio-economic terms, which is a rather non-geographical stance. Flows, frequency, connectivity, speed, loading, etc. are never considered. The structural dimension of transport networks often emphasized by Multi-Agent Systems is disregarded too.

5-Lines 159-160. To be removed? Is it still science?

6-Figure 2a. Please highlight PT areas. The legend must be clearer. Surprisingly, access to public transport services is seen through socio-economy and not population density, which is a very basic and fundamental criterion to introduce here. To be changed?

7-Line 354 “nearly perfect”. Clumsy. There are major discrepancies in the spatial distribution of PT stops.

8-Line 356 “lowest scoring”. Highest?

9-Fig. 3 and 5. Change colors to increase visual contrast (green), reverse color ranges (high rates, warm color)

10-Line 513. Language mistake.

11-Finally, I’m not sure providing basic information about GIS databases in annex is a good idea. I’d rather introduce here the structure of these databases (full description, including variables).

Reviewer 4 Report

This is an interesting geographical analysis of transport problems of Bogota, with respect to the United Nations’ SDG transport indicators.

The GIS analyses appear well done, but there are two concerns related to the validity of this research paper, which the authors need to address carefully before proceeding to resubmit their paper:

a)      None of the authors is from Colombia and neither appears any scientific paper in the literature references relating to transport in Colombia. Consequently, some reader might have grounds to question the validity of this research.

b)      There are several recent publications relating to transport analyses in Bogota. None of them has been examined by the authors of the present paper. For instance, the following:

A strategic and dynamic land-use   transport interaction model for Bogotá and its region

Guzman,   L.A.

2018

Transportmetrica B

pp. 1-19

Accessibility changes: Analysis of   the integrated public transport system of Bogotá

Guzman,   L.A., Oviedo,   D., Cardona,   R.

2018

Sustainability (Switzerland)

10(11),3958

Accessibility, affordability and   equity: Assessing ‘pro-poor’ public transport subsidies in Bogotá

Guzman,   L.A., Oviedo,   D.

2018

Transport Policy

68, pp.   37-51

Fare discrimination and daily   demand distribution in the BRT system in Bogotá

Guzman,   L.A., Moncada,   C.A., Gómez,   S.

2018

Public Transport

10(2), pp. 191-216

Democracy on the move? Bogotá’s   urban transport strategies and the access to the city

Vecchio,   G.

2017

City,   Territory and Architecture

4(1),15

The authors need to a) prove that their data are valid and b) they have considered the relevant literature carefully (it is not possible to publish on the transport system of Bogota without any scientific literature reference about it).

Also, the word Bogota needs to appear in the keywords (if not at the title also).

Reviewer 5 Report

The manuscript „Access or accessibility? A critique of the urban transport SDG indicator” is an interesting methodological paper aiming to critically assess the urban transport accessibility indicator used by the Sustainable Development Goals. It offers and tests on the example of Bogota two alternative measures more valid for the description of urban access to public transport. The paper is well structured and clearly written. I recommend the paper for publication in IJGI, yet I will provide a few suggestions to further improve the clarity of presentation, and add to the discussion of the results.

My first remark concerns the order of the presentation of the two new indicators proposed by authors. In the introduction, when describing the drawbaks of the 11.2 indicator as "supply" oriented, they first suggest the use of data on travel origins and destinations, and afterwards about actual travels. This order seems logical, as the first approach suggested by authors is methodologically closer to the 11.2 indicator, so logical order is: accessibility to transport infrastructure - accessibility of public transport connection - actual travels. I think the same order shuld be used in the next sections of the paper. Instead, the authors first describe the method and results of the use of actual travel data, and then about public transport connectivity.

Second, the above mentioned order also causes some confusion about the means of transport that the authors take into accout. Both 11.2 indicator and "potential accessibility" are based only on public transport, while the "actual travels" are based on all means of transport including private car trips. It would be interesting to see the actual travel data on public transport only, but the total trip data still presents valuable results. But it should be more clearly stated in the methodological part that private car trips are included there, unlike in other indicators.

Third, the authors might be more consistent in naming the indicators, it will ease the reading of the paper. When writing research questions, in question 3, the autors ask how "indicator 11.2" compares to "accessibility indicator" (one!), and in quesiton 4 they ask how to modify the indicator (11.2), even though in fact they do not finally suggest any modifications, but rather propose totally new indicators. The SDG indicator is sometimes referred to as "defined indicator", sometimes "indicator 11.2", and in the final table "Indicator 1 (current SDG)". The second indicator is "An indicator based on actual travel", "An indicator based on actual travel data" and "Accessibility based on actual travel", and similar applies to indicator 3 ("accessability rather than access", "potential accessibility"). Naming indicators consistently from the research questions until final comparison table, at least in the subsection titles and table headings, will make it the paper more readible.

Forth, in the discussion the authors might consider writing more on the "accessibility of accessibility data", which determines the possible application of the indicators proposed by the authors in other urban areas. I suppose one of the reasons behind the formulation of 11.2 indicator of SDG is the possibility to gather needed data (location of public transport stops and population data on high spatial resolution) in a large number of cities. Both authors' indicators require more sophisticated and less widely gather spatial data: on number of jobs and public transport network ("potential accessibility") and travel survey carried out on samples big enough to make it possible to subset respondents living in small reference areas ("actual travels"). I suppose the latter data is more difficult to obtain in most of the Global South cities.

Finally, there is a little typo in the very end of conclusions section: the authors appear to have left a part of the journal template text.

Round 2

Reviewer 2 Report

This is a much improved version of the previous paper, and the authors have addressed a number of the key points from the first review.  The specifications are more clear, and distinguishing between individual modes or mode combinations is helpful.  Including walking and bicycling trips also shows a comprehensive accessibility measurement, and it is interesting to note that including them improved the observed accessibility values.  The methods and results section are each much tighter, and I believe the paper is now ready for publication, pending a few minor edits.  There is no need to create a new section for sensitivity analysis on travel time, per the response to my initial comment.

1) Figure 3 a, change the representation of the number of jobs from a graduated symbol to a graduated color map.  Many of the block centroids and the graduated circles on top of them overlap.

2) Can the authors recommend some interventions that bridge the potential and actual travel indicator trips noted in Table 4, or explain why there isn't stronger correspondence between the potential and actual values?  This could be a function of policy, travel behavior, or types of jobs available.  To that end, it may be worth recommending certain job types (higher paying, lower paying, etc.) as an extension to this initial indicator, or perhaps they could be constructed for particular industries.

Reviewer 3 Report

I first want to thank the authors for their useful comments and changes to the text.

I discover that this paper was presented as an answer to a call for proposals about “Geo-Information and the Sustainable Development Goals (SDGs)”. I have accessed and read a number of already published articles and understand now more precisely the research initiative.

Many of my remarks have found an appropriate answer through this series of publications and through the answers provided by the authors.

1-It is clear now that this paper investigates one SDG indicator as a complement of other works already achieved. The consistency of the approach is ensured “by proxy”. My criticism about the sample validity of a single example to evaluate a global statistical indicator fades away in front of the actual methodological objective.

2-Miscalleneous remarks, not only about to the paper under review, but also about the special issue

-I hoped to find an analysis about the different “weights” of SDG indicators within the SDG panel. This perspective is essential to an appropriate understanding of what is at stake with this UN initiative. Many studies focus on a single indicator without investigating the overall purpose, biases or consistency of such a statistical “complex”.

-An assumed critical position supposes to discuss the objectives of these overall indicators elaborated by “technostructures” whose administrative and political positions are never neutral. This is why I relate the analysis about city transportation with geopolitics. The authors seemingly deny this relationship or ignore it, adopting a purely technical stance. Leaving aside the political dimension of SDGs is a mistake at least in my view.

-Lines 673-676 illustrate what I have called “some kind of ingenuity”, a persisting inclination in this paper. I do not think scientists should adopt the position of experts. Such a concluding sentence demonstrates that the authors intended to provide a technical answer to a technical question whose limits have been established by the UN itself. When scientists delegate their ability to define the boundaries of their hypotheses, demonstrations and conclusions to external institutions, they do not act anymore as scientists.

3-I still disagree about the colors used in Fig. 3 and 5: Inverting colors is necessary not for conceptual-connotation reasons but to comply with the nature of the human eye. This is about wavelength and vibration. Warm colors catch the attention of people whereas cold colors generate disregard. This is why important information must be figured with warm colors –typically the highest statistical values in a legend. Besides, light green and yellow are too close colors to be differentiated. Half of these maps escape visual scrutiny.

Reviewer 4 Report

Comments have been addressed to by the authors

Author Response

Thank you for your interest and contributions